# Rab11a mediates cell-cell spread and reassortment of influenza A virus genomes via tunneling nanotubes

**Ketaki Ganti**[1]*, **Julianna Han**[2¤], **Balaji Manicassamy**[3], **Anice C. Lowen**[1,4]*

**1** Department of Microbiology and Immunology, Emory University School of Medicine, Atlanta, Georgia, United States of America, **2** Department of Microbiology, University of Chicago, Chicago, Illinois, United States of America, **3** Department of Microbiology and Immunology, University of Iowa School of Medicine, Iowa City, Iowa, United States of America, **4** Emory-UGA Centers of Excellence for Influenza Research and Surveillance [CEIRS]

¤ Current address: Department of Integrative Structural and Computational Biology, The Scripps Research Institute, La Jolla, California, United States of America
* ketaki.ganti@emory.edu (KG); anice.lowen@emory.edu (ACL)

**Data Availability Statement:** All relevant data are within the manuscript and its Supporting Information files.

## Abstract

Influenza A virus [IAV] genomes comprise eight negative strand RNAs packaged into virions in the form of viral ribonucleoproteins [vRNPs]. Rab11a plays a crucial role in the transport of vRNPs from the nucleus to the plasma membrane via microtubules, allowing assembly and virus production. Here, we identify a novel function for Rab11a in the inter-cellular transport of IAV vRNPs using tunneling nanotubes [TNTs] as molecular highways. TNTs are F-Actin rich tubules that link the cytoplasm of nearby cells. In IAV-infected cells, Rab11a was visualized together with vRNPs in these actin-rich intercellular connections. To better examine viral spread via TNTs, we devised an infection system in which conventional, virion-mediated, spread was not possible. Namely, we generated HA-deficient reporter viruses which are unable to produce progeny virions but whose genomes can be replicated and trafficked. In this system, vRNP transfer to neighboring cells was observed and this transfer was found to be dependent on both actin and Rab11a. Generation of infectious virus via TNT transfer was confirmed using donor cells infected with HA-deficient virus and recipient cells stably expressing HA protein. Mixing donor cells infected with genetically distinct IAVs furthermore revealed the potential for Rab11a and TNTs to serve as a conduit for genome mixing and reassortment in IAV infections. These data therefore reveal a novel role for Rab11a in the IAV life cycle, which could have significant implications for within-host spread, genome reassortment and immune evasion.

## Author summary

Influenza A viruses infect epithelial cells of the upper and lower respiratory tract in humans. Infection is propagated by the generation of viral particles from infected cells, which disseminate within the tissue. Disseminating particles can encounter obstacles in the extracellular environment, including mucus, ciliary movement, antibody

**Funding:** This research project was supported in part by the Emory University Integrated Cellular Imaging Microscopy Core, supported by the Office of the Director, NIH (S10OD028673). The project was funded by NIH/NIAID Centers of Excellence in Influenza Research and Surveillance (CEIRS), contract number HHSN272201400004C (ACL; KG) and NIH R01AI127799 (ACL; KG). JH was partly supported by the NIH Molecular and Cellular Biology training program at The University of Chicago (T32GM007183); JH was partly supported by the NIH Diversity Supplement (R01AI123359-02S1). BM is supported by NIAID grants (R01AI123359 and R01AI127775). The funders had no role in study design, data collection and analysis, decision to publish, or preparation of the manuscript.

**Competing interests:** The authors have declared that no competing interests exist.

neutralization and uptake by phagocytic immune cells. An alternative mode of spread, which avoids these hazards, involves direct transport of viral components between cells. This cell-cell spread of infection is not a well understood process. In this study we demonstrate that the host factor Rab11a mediates the transport of viral genomes in the cell-cell spread of infection. Rab11a is already known to play a pro-viral role in the transport of viral genomes to the plasma membrane for assembly into virus particles. Here, we see that this same transport mechanism is co-opted for direct cell-cell spread through cellular connections called tunneling nanotubes. We show that complexes of Rab11a and viral components can be trafficked across tunneling nanotubes, transmitting infection without the formation of virus particles. Importantly, this route of spread often seeds viral genomes from multiple donor cells into recipient cells, which in turn increases viral genetic diversity.

## Introduction

Influenza A virus[IAV] genomes are composed of eight RNA segments that are packaged into the virion in the form of viral ribonucleoproteins [vRNPs], which contain viral nucleoprotein [NP] as well as the polymerase complex [PB2, PB1 and PA] [1]. Influenza genome packaging mechanisms have been studied extensively and, although there are a lot of unknowns, it has been demonstratively shown that the host cell protein Rab11a is crucial for the trafficking of newly synthesized vRNPs after they exit the nucleus to the site of assembly at the plasma membrane [2,3]. Rab11a is a small GTPase that has multiple roles in the host cell, including a pivotal role in retrograde transport of cargoon recycling endosomes [4,5]. The intracellular transport of vRNP-Rab11a complexes is thought to occur via the microtubule network with the help of dynein motors [6–11]. There are, however, conflicting observations about the impact of microtubule disruptionon the viral lifecycle, with results ranging from no detectable effect [6] to an attenuation of viral progeny production [8]. Our prior work revealed that loss of Rab11a reduces infectious viral titers, most likely due to a defect in the packaging of vRNPs, leading to the formation of incomplete viral particles [12]. Taken together, these data demonstrate the importance of an intact microtubule network as well as Rab11a in the IAV life cycle.

Tunneling nanotubes [TNTs] are F-Actin rich cellular connections that are formed between two or more cells [13]. These connections can be formed over long distances and provide cytoplasmic connectivity between the cells, allowing for exchange of materials including organelles, nutrients, and membrane vesicles [14–16]. Many viruses including HIV [17–19], herpesviruses [20] and IAVs [21,22] have been shown to utilize these TNTs for cell-cell spread. Previous work has shown that IAV spread via TNTs proceeds in the presence of neutralizing antibodies or antivirals such as oseltamivir [21,22]. This mode of infection does not depend on the formation of viral particles, thus allowing for the assembly stage of the lifecycle to be bypassed. Although the use of TNTs by IAVs has been demonstrated, the exact mechanism is unclear.

In this study, we show that Rab11a mediates the transport of IAV vRNPs and proteins through TNTs, as evidenced by Rab11a co-localization with viral components in TNTs and the disruption of this transport between Rab11a knock out cells. This system was observed to be functional in multiple host cell backgrounds and virus strains. Using HA deficient viruses, we confirm that transport of viral components through TNTs can seed productive infection in recipient cells. In the context of viral co-infection, we find direct cell-cell spread often seeds viral genomes from multiple donor cells into recipient cells, thus romoting genome mixing and reassortment. Finally, our data suggest that, at least in the case of IAV infection, TNTs

access the cytosol of both connected cells and allow bi-directional movement of cargo. Taken together, these findingsdemonstrate a novel and crucial role for Rab11a in the trafficking of IAV genomes via tunneling nanotubes and extend mechanistic understanding of this unconventional mode of viral dissemination.

## Results

### IAV vRNPs associate with Rab11a within F-Actin rich TNTs

Upon nuclear exit, IAV vRNPs bind to Rab11a via PB2, allowing their transport to the plasma membrane for assembly [8,23,24]. We hypothesized that vRNP-Rab11a complexes could also be routed to F-Actin rich intercellular connections called tunneling nanotubes [TNTs] and could seed new infections by direct transport through TNTs. To test this hypothesis, we visualized Rab11a, F-Actin and viral nucleoprotein [NP]—as a marker for vRNPs—in MDCK cells infected with either influenza A/Netherlands/602/2009 [NL09; pH1N1] or A/Panama/2007/99 [P99; H3N2] virus. NP and Rab11a were seen to co-localize in a perinuclear compartment, as has been shown previously [2,9]. In addition, co-localization of these components was observed within the F-Actin rich TNTs connecting infected and uninfected cells [Figs 1A and S1]. This observation suggests that there are at least two functional pathways for the trafficking of vRNP-Rab11a complexes post nuclear exit: the canonical assembly pathway and the TNT-mediated genome transfer pathway.

To further corroborate the role of Rab11a in the transport of vRNPs across TNTs, we used Rab11a knockout [KO]A549 cells generated by CRISPR/Cas9 [12] and wild type [WT]A549 cells as a control. As before, cells were infected with either NL09 or P99 viruses and then stained for NP, Rab11a and F-Actin. WT cells showed co-localization of NP and Rab11a in the perinuclear region and within TNTs [Figs 1B and S2]. Conversely, Rab11a KO cells did not show NP staining within the TNTs, indicating that Rab11a drives the transfer of vRNPs through TNTs [Figs 1C and S3].

We used super resolution Stimulated Emission Depletion [STED] microscopy to analyze the association of Rab11a and vRNPs within TNTs in more detail. STED microscopy overcomes the diffraction resolution limit of confocal microscopy and allows for imaging with up to 30nm resolution [25,26]. A549 WT or Rab11a KO cells were infected with NL09 viruses and stained for NP, Rab11a and F-Actin. Using 3 color STED imaging of TNTs, we observed punctate Rab11a and NP staining in close proximity within the TNTs in WT cells. In the case of the Rab11a KO cells, the NP staining was relatively diffuse in the cytoplasm and could not be observed within the TNTs [Fig 2A]. Co-localization of Rab11a and NP within TNTs was analyzed quantitatively for both WT and KO cells. A resolution of 41.67 nm for the Rab11a-NP staining was achieved, giving a high degree of confidence in the co-localization quantitation. Of note, Rab11a was previously determined to interact with vRNPs via PB2 [24] and not with NP, the target of our vRNP staining. The observed colocalization co-efficient of 0.9 in the WT cells indicates that the majority of the NP puncta within TNTs coincide with Rab11a puncta [Fig 2B]. These data indicate that Rab11a and vRNPs are indeed interacting with each other within TNTs and are likely trafficked as a complex.

To evaluate the frequency with which Rab11a accesses TNTs, we quantified the percentage of TNTs that were NP+Rab11a+, NP+Rab11a- and NP-Rab11a+ in uninfected and NL09 infected A549 WT cells. Rab11a was routinely detected within the TNTs of uninfected cells, indicating that Rab11a+ recycling endosomes are trafficked within TNTs under normal conditions [Fig 2C]. In infected cells, a majority of TNTs carried NP+Rab11a+ puncta, indicating that transport of vRNPs into TNTs is a common feature of the viral life cycle [Fig 2C].

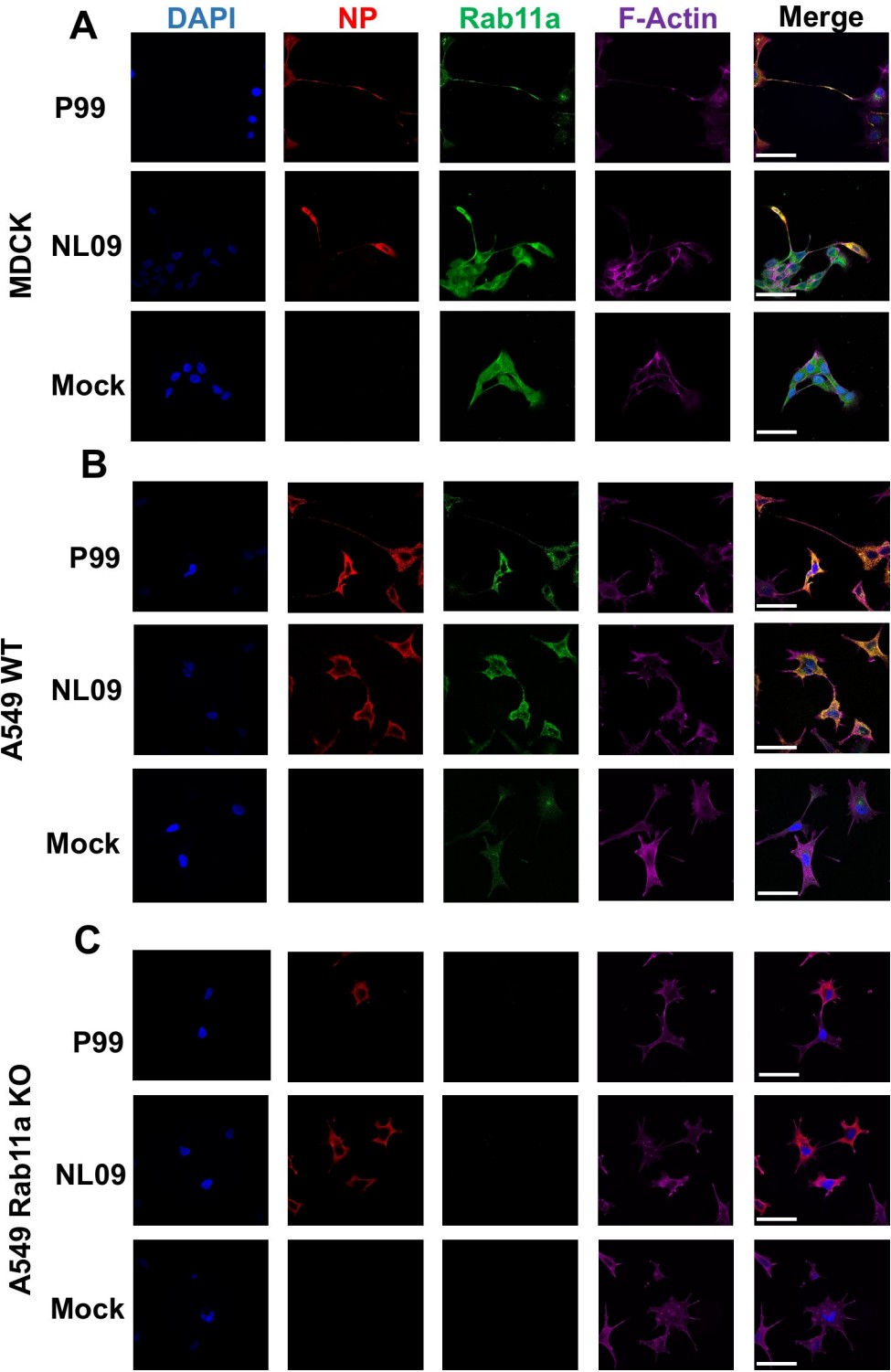

**Fig 1. IAV vRNPs associate with Rab11a in F-Actin rich TNTs.** [A] MDCK cells,[B] A549 WT and [C]A549 Rab11a KO cells were mock-infected or infected with NL09 or P99 viruses. Cells were stained for DAPI [blue], NP [red], Rab11a [green] and F-Actin [pink]. Representative images are shown withadditional images in S1–S3 Figs. Scale bar is 20μm for all images.

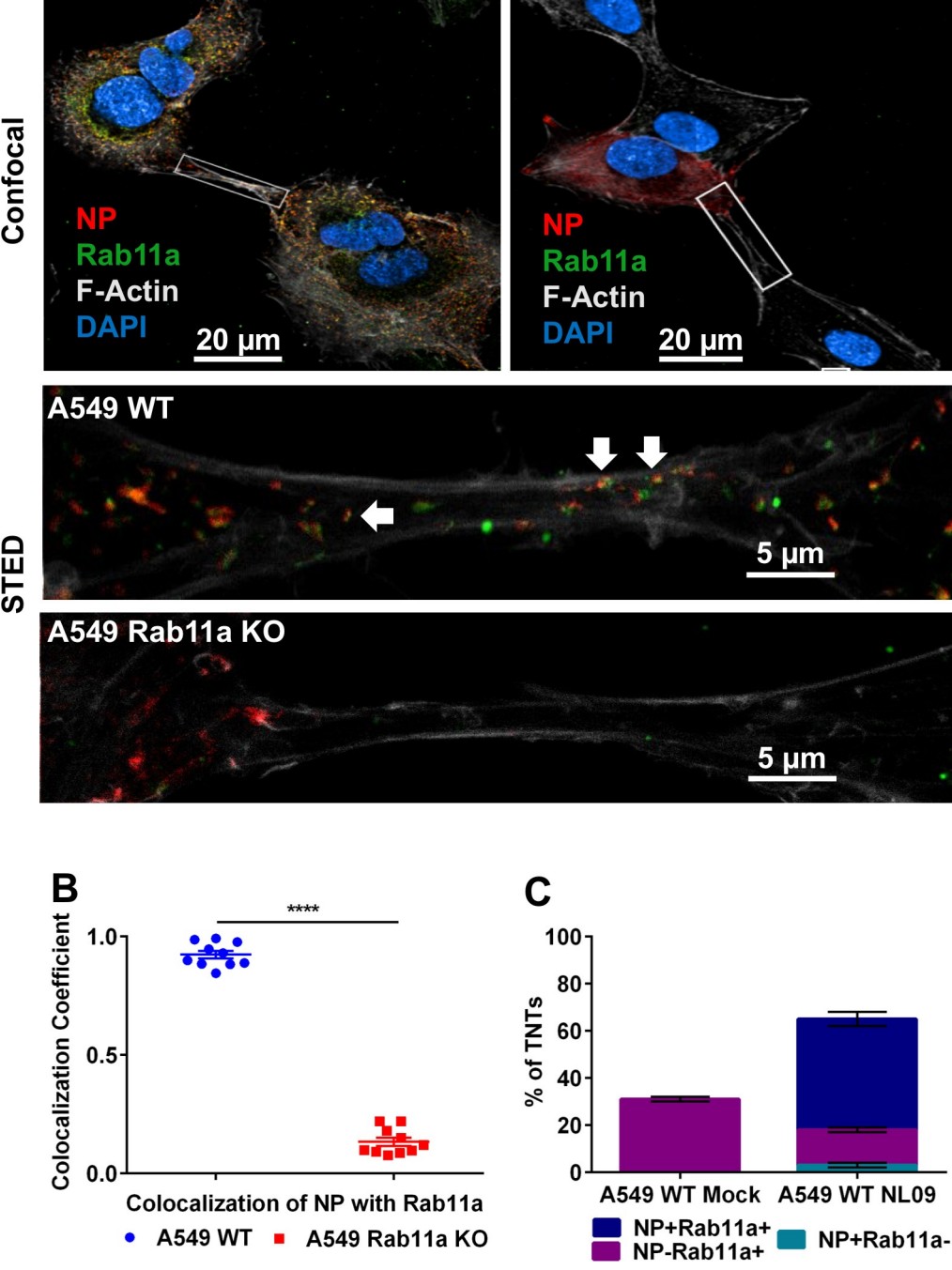

**Fig 2. IAV vRNPs co-localize with Rab11a within TNTs.** [A]A549 WT and Rab11a KO cells were infected with NL09 viruses. Cells were stained for DAPI [blue], NP [red], Rab11a [green] and F-Actin [grey]. Representative confocal images are shown with the high-resolution STED images depicted in the insets. The arrows depict the co-localized puncta of NP and Rab11a. [B] Co-localization coefficient of NP and Rab11a within TNTs from A549 WT and Rab11a KO cells [n = 10 TNTs]. Significance of differences in co-localization between the WT and KO cells was tested using a two tailed unpaired t-test [**** P-value <0.0001]. [C] A549 WT cells were mock infected or infected with NL09 viruses and the percentage of TNTs formed between 50 cell pairs were counted manually as NP+Rab11a+ [blue], NP-Rab11a+ [magenta] and NP+Rab11a- [teal]. Error bars represent the SEM of two biological replicates.

## Rab11a does not modulate TNT formation

IAV infection increases the number of TNTs formed between cells [21,22]. To test whetherthe loss of Rab11a had an impact on TNT formation, we counted the number of TNTs formed between cells in WT and Rab11a KO cells in the presence and absence of NL09 infection. Consistent with prior work, a significant increase in the number of TNTs formed upon IAV infection was observed in both WT and KO cells [Fig 3]. Conversely, we saw no significant difference between WT and KO cells in the number of TNTs formed, either in the context of infection or mock infection [Fig 3]. The observation that loss of Rab11a does not significantly impact TNT formation suggests that the reliance of vRNP trafficking through TNTs on Rab11a arises through the observed interaction between these components.

## Disruption of actin or loss of Rab11a significantly attenuates direct cell-cell transmission of infection

Since we observed the presence of vRNP-Rab11a complexes within TNTs, we next tested whether the loss of either the TNTs or Rab11a influences the cell-cell spread of IAV infection. Previously, neutralizing antibodies or neuraminidase inhibitors have been used to block conventional viral infection and allow examination of IAV protein and RNA transport via TNTs [21,22]. Although these methods are effective in abrogating conventional spread, we wanted to fully eliminate the generation of viral progeny to define the role of Rab11a and TNTs in IAV

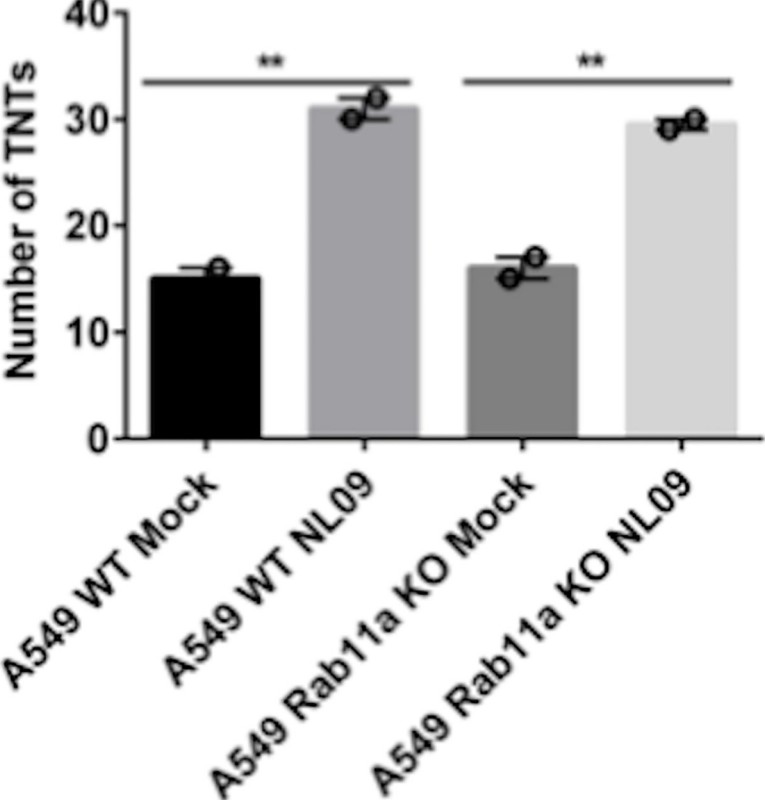

**Fig 3. Rab11a does not modulate TNT formation.** A549 WT and Rab11a KO cells were mock infected or infected with NL09 viruses and the total number of TNTs formed between 50 cell pairs were counted manually. Significance of differences in the number of TNTs between mock and NL09 infected groups was tested using 1-way ANOVA [** P-value <0.01]. Error bars represent the SEM of two biological replicates.

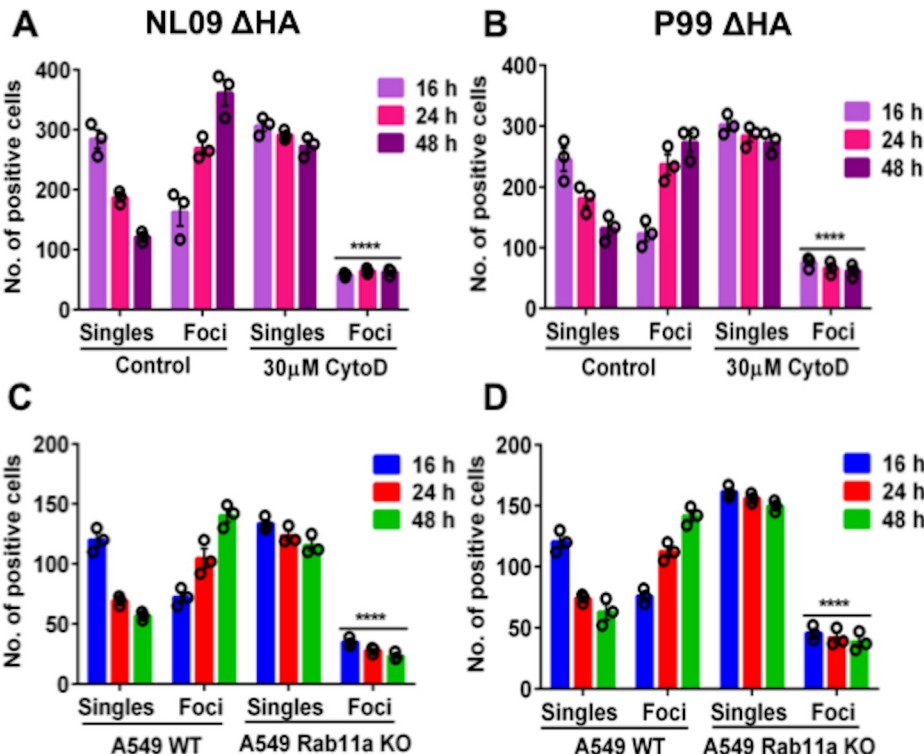

**Fig 4. Disruption of actin or loss of Rab11a abrogates direct cell-cell transmission of infection.** MDCK cells infected with NL09 ΔHA Venus WT [A] or P99 ΔHA Venus WT [B] at a MOI of 0.5 were counted as single infected cells or foci of infected cells. Significance of differences in the number of infected foci between the control and Cytochalasin D treated groups was tested using 2-way ANOVA with Bonferroni's correction for multiple comparisons [**** P-value <0.0001]. Error bars represent the standard error of three biological replicates. A549 WT or A549 Rab11a KO cells infected with NL09 ΔHA Venus WT [C] or P99 ΔHA Venus WT [D] at a MOI of 0.5 were counted as single infected cells or foci of infected cells. Significance of differences in the number of infected foci between cell types was tested using 2-way ANOVA with Bonferroni's correction for multiple comparisons [**** P-value <0.0001]. Error bars represent the standard error of three biological replicates.

genome transfer more clearly. To this end, we rescued recombinant viruses in the NL09 and P99 strain backgrounds that lack the HA gene but instead contain either mVenus [NL09 ΔHAVenus; P99 ΔHAVenus] or mScarlet [NL09 ΔHAScarlet] fluorescent reporter ORFs flanked by HA packaging signals. These HA deficient reporter viruses are infection competent but are unable to produce progeny in the absence of a HA complementing cell line. Therefore, these viruses are excellent tools to study the cell-cell spread of IAV infection via TNTs.

To analyze the role of F-actin in the cell-cell spread of viral genomes, MDCK cells were infected with either NL09 ΔHAVenus or P99 ΔHA Venus viruses in the presence or absence of Cytochalasin D, which is a potent inhibitor of actin polymerization and disrupts TNTs[21,22]. Cytochalasin D was added 2h post internalization. mVenus positive cells were counted at 16, 24, and 48 h post-infection [p.i.]and binned into one of two categories: single cells or foci comprising > = 2 contiguous, positive cells. We hypothesized that the disruption of TNTs by Cytochalasin D would severely limit the spread of IAV genomes from infected cells, preventing the formation of infected foci. As shown in Fig 4A and 4B, there was a significant reduction in the number of infected foci in the Cytochalasin D treated cells compared to the untreated controls in both the NL09 ΔHA Venus and P99 ΔHA Venus infected cells. These data confirm that intact TNTs are required for direct cell-cell spread of IAV genetic material.

Next, we analyzed the role of Rab11a in direct cell-cell spread of IAV. To do this, A549 WT and Rab11a KO cells were infected with either NL09 ΔHA Venus or P99 ΔHA Venus viruses. Venus positive cells were counted at 16, 24, and 48 h p.i. and categorized based on their presence as single cells or within foci at each time point. If Rab11a directs transport of vRNPs across TNTs, the loss of Rab11a would be expected to reduce the cell-cell spread of IAV genetic material. As shown in Fig 4C and 4D, this was indeed the case. In contrast to WT controls, the number of infected foci did not increase over time in the Rab11 KO cells. These data provide further evidence for the role of Rab11a in this alternate infection pathway.

## Virion-independent genome transfer leads to productive infection by an actin-dependent mechanism

To assess whether all eight genome segments can be transported via TNTs leading to the production of infectious progeny, we performed a co-culture experiment using MDCK cells and a MDCK-derived cell line which expresses the HA ofinfluenza A/WSN/33 [H1N1] virus on the cell surface [MDCK WSN HA cells]. When infected with an HA deficient reporter virus, these cells provide the HA protein required for the generation of infectious virus particles. For subsequent analysis of co-infection via TNT/Rab11a mediated genome transfer, these experiments were set up using two IAV strains, NL09 ΔHAVenus WT and NL09 ΔHAScarlet VAR viruses. In addition to carrying differing reporter genes, these viruses differ in the presence of a silent mutation in each segment of the VAR virus, which acts as a genetic tag [27]. Neither of these differences is important for the purposes of the present analysis.

As outlined in Fig 5A, separate dishes of MDCK cells were singly infected with either NL09 ΔHAVenus WT or NL09 ΔHAScarlet VAR virus. After infection for 2 hours, the cells were acid washed to remove residual inoculum and then trypsinized to make a cell slurry. MDCK cells infected with NL09 ΔHAVenus WT and NL09 ΔHAScarlet VAR were mixed with naïve MDCK WSN HA cells in the ratio 1:1:2. The cell mixture was plated in medium containing trypsin [to allow activation of HA] and ammonium chloride [to prevent secondary infections mediated by virus particles]. If all eight segments of the viral genome can be transported across TNTs to a conducive cell, which in this case must be a MDCK WSN HA cell, then the recipient cell will produce virus particles. To detect any such progeny viruses produced, supernatant was collected at 0, 24, 48 and 72 h post mixing and plaque assays were performed on MDCK WSN HA cells.

To evaluate the role of TNTs in cell-cell spread of infection, cells were treated with either vehicle or 30 μM Cytochalasin D, which disrupts F-Actin. As can be seen from Fig 5B, infectious virus was detected in the vehicle treated control cells, but not in the Cytochalasin D treated cells. Virus production in vehicle treated cells demonstrates the transfer of the full complement of IAV genome segments from infected cells, which lack HA protein and cannot produce virions, to cells which express complementing HA protein. A lack of virus production in Cytochalasin D treated cells indicates that this transfer was F-Actin-dependent, strongly implicating TNTs. Comparing the two MOIs tested[2.5 and 25 PFU/cell], a dose dependence was observed at 24 h, which is most likely due to the increased probability of an infected cell making a connection with a naïve MDCK WSN HA cell at higher MOI.

To assess if actin depolymerization was impairing vRNP transport upstream of TNT transfer, we tested the effect of Cytochalasin D on conventional virus production in the context of multicycle replication. No difference in viral yield from vehicle treated or Cytochalasin D treated cells was detected [Fig 5C]. Taken together, these data indicate that actin depolymerization specifically abrogates TNT mediated transfer of viral genomes but has no impact on conventional viral assembly.

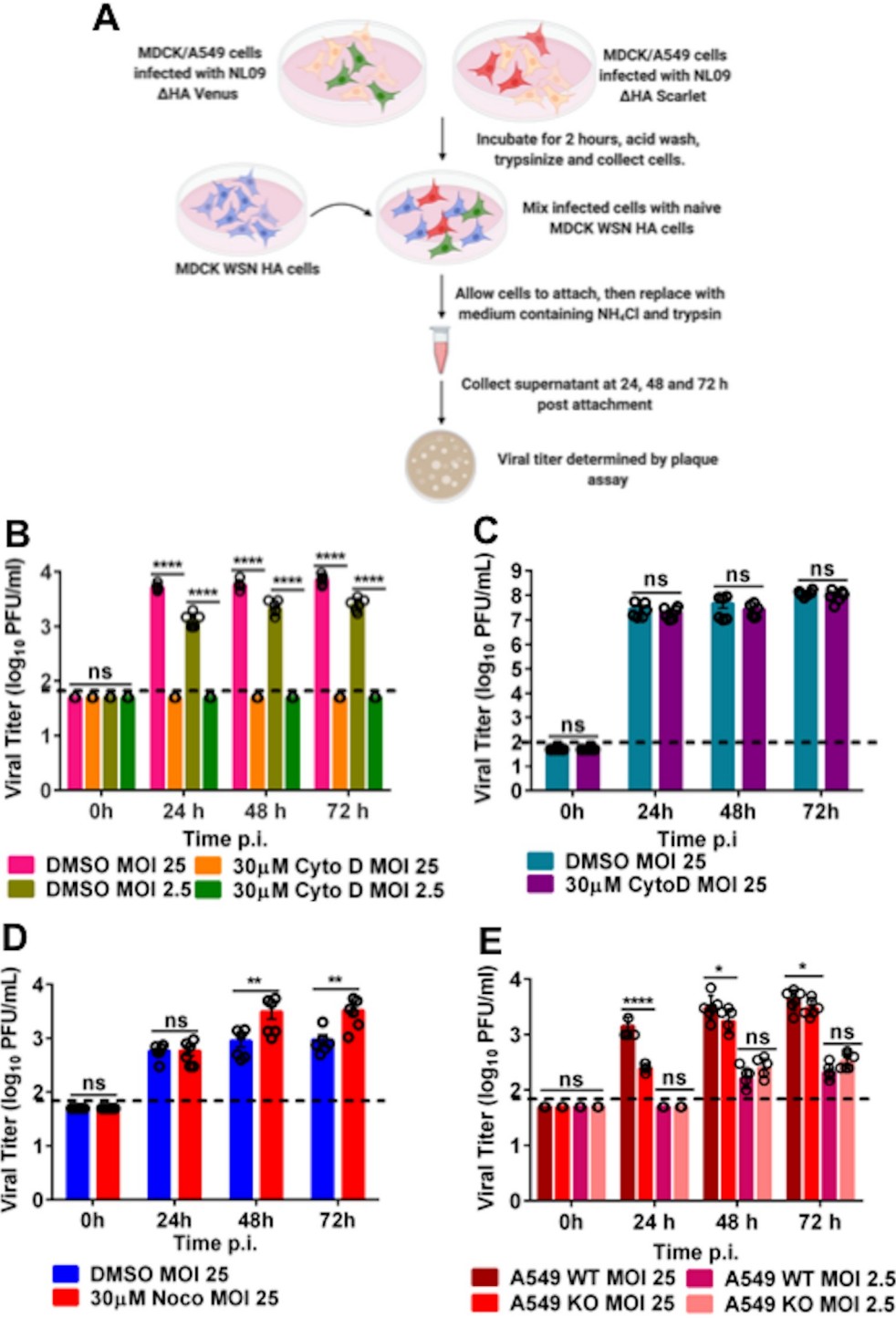

**Fig 5. Virion-independent genome transfer leads to productive infection by an actin-dependent mechanism.** [A] Experimental workflow for MDCK, A549 WT or A549 Rab11a KO infection and co-culture with MDCK WSN HA cells [generated via BioRender.com]. Plotted is the infectious virus yield from co-culture of MDCK WSN HA cells with MDCK cells infected with NL09 ΔHA Venus WT and NL09 ΔHAScarlet VAR viruses either treated with Cytochalasin D [B] or Nocodazole [D]. [C] Infectious virus yield from MDCK cells infected with NL09 viruses and treated with Cytochalasin D. [E] Infectious virus yield from co-culture of MDCK WSN HA cells with A549 WT or Rab11a KO cells infected with either NL09 ΔHA Venus WT or NL09 ΔHAScarlet VAR viruses. Significance of differences between the control and Cytochalasin D or Nocodazole treated cells or between WT and KO cells was tested using 2-way ANOVA with Bonferroni's correction for multiple comparisons [* P-value <0.1; ** P-value <0.01; *** P-value <0.001; **** P-

value <0.0001; ns = not significant]. Error bars represent the SEM of two biological replicates,each comprising three replicate infections[black circles]. The dotted line represents the limit of detection of the plaque assay.

To test if microtubules play a role in TNT mediated transport of IAV genomes, we performed the co-culture assay as described above with vehicle treated and 30 μM Nocodazole, which depolymerizes microtubules. We detected infectious virus in both control and nocodazole treated cells [Fig 5D], indicating that microtubules most likely did not play a role in the trafficking of vRNPs through TNTs.

Finally, to analyze the effect of the loss of Rab11a on the production of infectious progeny, we co-cultured either A549 WT or A549 Rab11a KO cells infected with the NL09 ΔHAVenus WT or NL09 ΔHAScarlet VAR viruses with the MDCK WSN HA complementing line as described above. Supernatant was collected at 24, 48 and 72 h post mixing and plaque assays were performed on MDCK WSN HA cells. As can be seen from Fig 5E, infectious virus was detected from both A549 WT and Rab11a KO cells, with a marginal difference in titers at 48 and 72 hpi. Since this observation was incongruent with our previous data demonstrating the importance of Rab11a in the transport of vRNPs, we hypothesized the transfer of viral genomes from the Rab11a KO cells was occurring via Rab11a that originates in the MDCK WSN HA cells. If correct, this observation would indicate that TNTs are open ended and allow for bi-directional movement of cargo.

## TNTs likely allow bidirectional shuttling of Rab11a between cells

To analyze if Rab11a could shuttle form one cell to another in a bi-directional manner, we used A549 WT and Rab11a KO cells in combination. Briefly, A549 Rab11a KO cells were infected with NL09 WT virus. The infected cells were then mixed in a 1:1 ratio with uninfected A549 WT cells which were pre-stained with CellTracker Blue dye. The mixed cells were stained for NP, Rab11a and F-Actin at 24h post-mixing and imaged using STED microscopy. We observed NP and Rab11a within the TNTs connecting the infected KO cells and the uninfected WT cells[Fig 6A]. Co-localization analysis again showed Rab11a-NP colocalization within TNTs in this KO-WT co-culture system [Fig 6B], although the coefficient was lower than that seen in a fully Rab11a competent system [Fig 2B]. Since the WT cell is the only source of Rab11a in the KO-WT co-culture, our data show that Rab11a traveling from an uninfected cell to an infected cell can pick up vRNPs. The production of viral progeny observed in this system further indicates that this Rab11a from the originally uninfected cell can then transport vRNPs back through the TNTs, mediating infection. Our data suggest that TNTs formed in the context of IAV infection are most likely open ended and bi-directional.

## TNTs serve as conduits for genome mixing and reassortment

Since we observed that infectious progeny could be generated via Rab11a-mediated genome transfer through TNTs, we hypothesized that this process could also mediate co-infection and therefore reassortment. In particular, reassortment would be expected if differentially infected donor cells connect to the same recipient cell. To test this hypothesis, the genotypes of virus produced from the co-cultures described in Fig 5E were evaluated. In these experiments, cells infected with NL09 ΔHA Venus WT virus were mixed with cells infected withNL09 ΔHA Scarlet VAR virus and these infected cells were in turn mixed with MDCK WSN HA cells; thus, co-infections could occur if WT infected and VAR infected cells each formed connections with the same HA-expressing recipient cell and a full complement of IAV segments was reconstituted therein. The silent mutations differentiating each of the non-HA gene segments of the

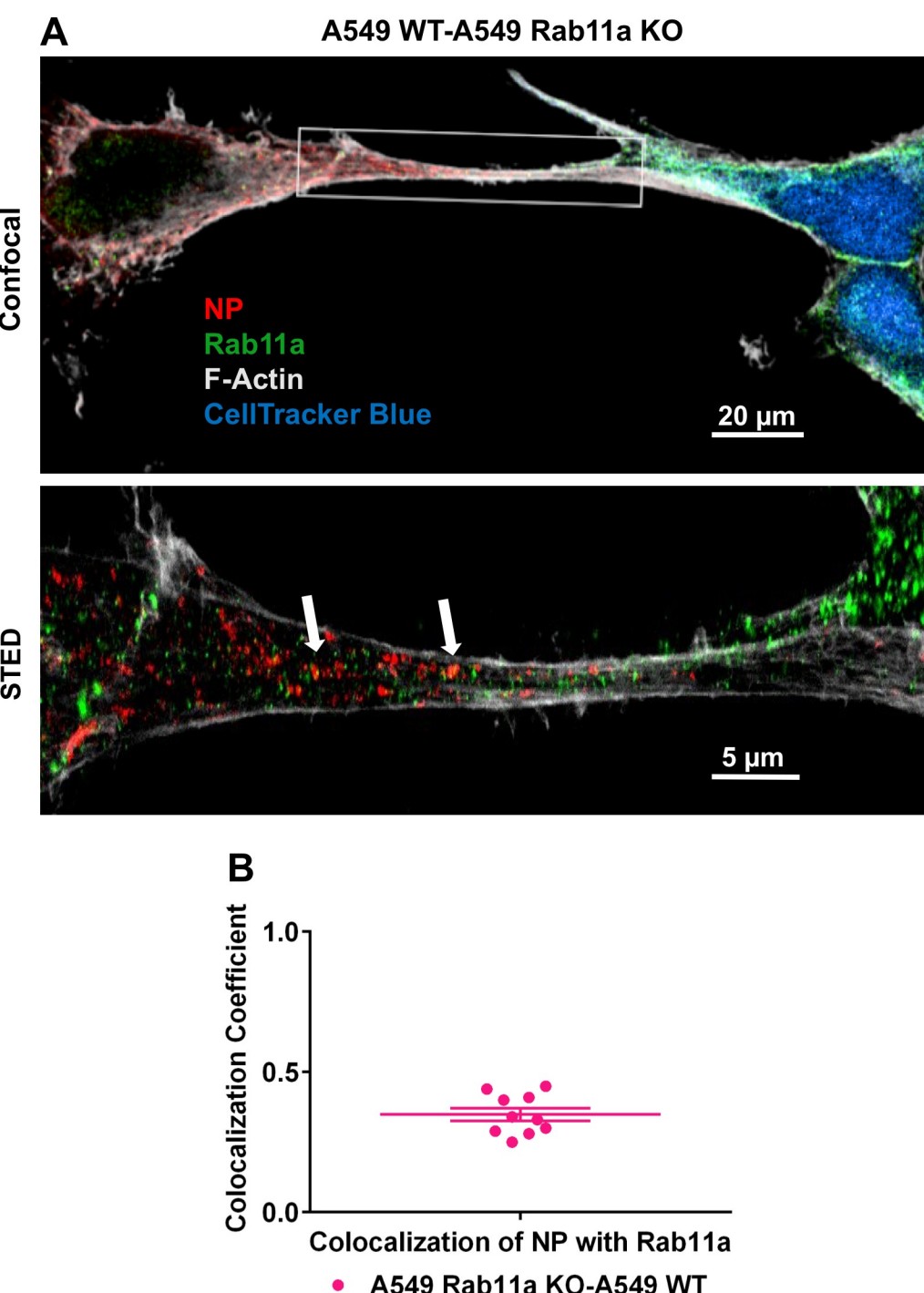

**Fig 6. TNTs allow for bidirectional shuttling of Rab11a between cells.** [A]A549 Rab11a KO cells were infected with NL09 viruses and mixed with A549 WT cells pre-stained with CellTracker Blue CMAC dye [blue]. Cells were stained for NP [red], Rab11a [green] and F-Actin [grey]. Representative confocal image is shown with the high-resolution STED images depicted in the inset. [B] Co-localization coefficient of NP and Rab11a within TNTs formed between A549 WT and Rab11a KO cells [n = 10 TNTs]. Error bars represent the SEM of two biological replicates with two technical replicates each.

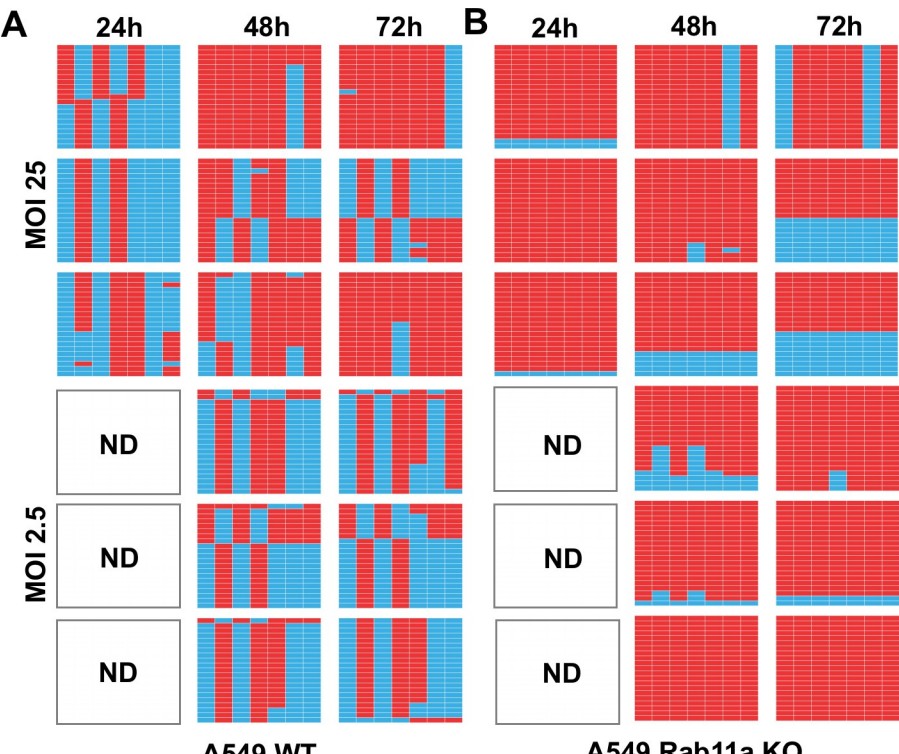

**Fig 7. TNTs serve as conduits for genome mixing and reassortment in a Rab11a dependent manner.** Genotypes of clonal viral isolates collected from the culture medium of NL09 ΔHA Venus WT and NL09 ΔHA Scarlet VAR virus infected A549 WTcells [A]or A549 Rab11a KOcells [B] co-cultured with MDCK WSN HA cells. Three technicalreplicate co-cultures per condition were sampled serially at the time points indicated and 21 plaque isolates were analyzed per sample. The origin of the gene segments, represented by the columns in each table, is denoted by the colored boxes [blue = WT and red = VAR]. The segments are in order PB2, PB1, PA, NP, NA, M and NS moving from left to right. The white panels indicate samples where no plaques were detected [ND = not detected].

WT and VAR viruses allow the parental origin of segments to be identified. Thus, to evaluate reassortment, plaque clones were isolated from co-culture supernatants and the genotype of each was determined. The results show that viruses generated from MDCK WSN HA cells mixed with infected A549 WT cells were predominantly reassortant under all conditions evaluated [Fig 7A]. In contrast, when MDCK WSN HA cells were mixed with infected A549 Rab11 KO cells, parental viruses typically dominated [Fig 7B]. Thus, in a Rab11a-sufficient system, intercellular transfer of IAV vRNPs through TNTs readily yielded reassortants, indicating that TNTs are forming a network rather than pairwise connections between cells. When Rab11a was absent from infected donor cells, however, reassortants were rarely observed. Since Rab11a knock out does not impact the number of TNTs formed [Fig 3], this relative lack of reassortants suggests that vRNP transport through TNTs is less efficient when Rab11a is absent from donor cells.

We note that, in both data sets shown in Fig 7, richness of viral genotypes was low, with at most four distinct gene constellations detected in each sample of 21 plaque isolates. This observation suggests that very few cells are producing most of the progeny virus in this experimental system, and that each producer cell is releasing virus with only one or a small number of genotypes. In turn, this suggests that MDCK WSN HA cells that receive a full complement of IAV vRNPs do not tend to receive multiple copies of a given segment. Although low in both culture systems, richness was significantly higher in the samples derived from A549 WT cells

compared to those from A549 Rab11a KO cells, with 2.8 and 1.9 unique genotypes detected on average, respectively [p = 0.019, t-test]. This difference is consistent with less efficient vRNP transfer when donor cells lack Rab11a.

## Discussion

Our data reveal a novel role for the host GTPase Rab11a in the trafficking of IAV genomes via tunneling nanotubes. We decisively show that productive infection can be mediated through this direct cell-to-cell route and find evidence that Rab11a can move through TNTs in a bidirectional manner to mediate IAV genome transfer. In the context of mixed infections, we furthermore find that TNT/Rab11a-mediated transfer readily leads to cellular coinfection and reassortment.

The trafficking of IAV genomes is a complex and poorly understood process. Although it is known that newly synthesized vRNPs form transient complexes with active Rab11a post nuclear exit and are trafficked to the plasma membrane for assembly on microtubule structures [6–9], the fate of these complexes is not completely elucidated. Here we examined the potential for Rab11a-vRNP complexes to be trafficked through TNTs to neighboring cells. Tunneling nanotubes [TNTs] are F-Actin based cytoplasmic connections that are utilized for long distance communication and have been shown to have a role in the IAV life cycle [21,22]. TNTs can be used to transport vesicular cargo [14,28–30], so we posed the question of whether the Rab11a-vRNP vesicular complexes could be re-routed to these structures. We show that Rab11a and vRNPs co-localize within TNTs in multiple cell types, with near complete concordance of NP with Rab11a in high resolution STED images of TNTs. Loss of Rab11a leads to severely reduced detection of NP within the TNTs and more dispersed NP localization within the cytoplasm. These observations strongly suggest that Rab11a-vRNP complexes are transported within TNTs.

TNTs are mainly composed of F-Actin and the transport of organelles through TNTs requires myosin motor activity on actin filaments [11,23–25]. Since Rab11a can utilize both dynein motors, which drive microtubule movement [6,26,27], and myosin motors, which drive actin dynamics [28–30], the observation that Rab11a mediates transport through TNTs raises the question of which motor proteins are involved. Studies to date on IAV infection have mainly focused on the role of Rab11a and microtubules. Further studies are needed to determine whether the same transport mechanism is active within TNTs and, conversely, whether Rab11a-actin dynamics may function in vRNP transport both within and between cells. Although Rab11a has been shown to be important for trafficking and efficient assembly of IAV genomes [2,8,12], we have shown that the loss of Rab11a does not completely abrogate genome assembly and viral particle release, indicating that there may exist an alternative pathway for canonical assembly of virions [12].

We observed that infectious progeny could be recovered from co-cultures of infected Rab11a KO A549 cells with MDCK WSN HA cells, which was incongruous with our previous observation that Rab11a KO abrogated the cell-cell transmission of infection. We were able to resolve this paradox by utilizing high-resolution STED imaging to visualize the transport of Rab11a from uninfected A549 WT cells to infected A549 Rab11a KO cells, where vRNPs could be picked up and trafficked through TNTs. TNTs can be formed in multiple ways- single ended, open ended or closed—and therefore support varying modes of transport [13,16,31]. The generation of progeny virions in the Rab11a KO co-culture is likely due to the formation of open ended, bi-directional TNTs that allow Rab11a from the HA-expressing producer cell to shuttle to and from infected KO cells where it could pick up vRNPs. This process seems to be inefficient, however, as evidenced by the low rate of reassortment observed when infected

cells do not encode Rab11a. Bi-directional transfer of organelles such as mitochondria through TNTs have been observed in various cell types, including lung mesothelioma cells [32]. It is also possible that there exists another host factor that can mediate vRNP transport through TNTs in the absence of Rab11a, albeit inefficiently. The possibility of bidirectional trafficking of vRNPs between cells, or potential novel host factors involved in vRNP transport, opens hitherto unexplored avenues of viral infection.

Our data revealing that coinfection and reassortment can occur through TNT transfer of vRNPs between cells raise new questions about the processes driving IAV genetic exchange. The prevalence of reassortants produced via TNT transfer from Rab11a+ cells indicates that vRNPs may be trafficked individually or as subgroups and not as a constellation of 8 segments. This process would then seed incomplete viral genomes into recipient cells, which require complementation to allow the production of progeny viruses. Owing to this reliance on complementation, incomplete viral genomes are known to augment reassortment[33–35]. Our data suggest that both seeding and complementation of incomplete viral genomes can occur via TNT transfer of vRNPs. In the presence of a conventional viral infection system, co-infection with multiple virions is thought to be the *modus operandi* of IAV reassortment, where reassortment efficiency is a function of the dose and relative timing of two infections, as well as levels of incomplete viral genomes [27,33]. It will be interesting to determine whether TNT-mediated co-infection is also sensitive to dose and timing. More broadly, further work is needed to tease out the extent to which TNT mediated reassortment occurs alongside conventional modes of reassortment.

The human airway is composed of multiple cell types, including polarized epithelial cells in a tightly packed environment [36]. It is yet unclear what role TNTs may play in the normal homeostasis of the airway, but TNTs have been demonstrated to mediate the transfer of cargo and organelles in various solid tumors of the lung, including adenocarcinomas and mesotheliomas [32]. TNT mediated transfer of organelles and other components including viruses has been observed between heterotypic cells, mostly involving immune cells such as macrophages [37–40]. It is therefore likely that IAV infection of the airway epithelium or potentially immune cells can spread via TNTs *in vivo*. The extent to which TNT mediated spread acts in conjunction with particle-based transmission of infection within the host is as yet unexplored. IAV spread through TNTs may be particularly important in the evasion of antibodies, and other antiviral factors that act directly on extracellular virions, in a manner that does not depend on the generation of escape mutants. Additional routes of HA-independent direct cell-cell spread of infection are possible. It has been recently shown that cell-cell spread of H5N1 IAVs can occur via trogocytosis, in which there is an actin dependent exchange of the plasma membrane and its associated molecules between conjugated cells [41]. Direct cell-cell spread of human metapneumovirus occurs via reorganization of the actin cytoskeleton [42], while measles virus infection of neighboring cells can occur through tight junctions [43,44]; whether IAV exploits these two routes of spread remains unclear. Direct cell-cell spread may make an important contribution to spatial structure of infection within the host, leading to more localized spread and limiting mixing among *de novo* variants [45]. To further investigate these potential implications, an exciting prospect for future work is the development of *ex vivo* and *in vivo* models for the study and visualization of TNT mediated cell-cell spread.

In summary, our data show a novel role for Rab11a in the IAV life cycle, where it can mediate vesicular transport of vRNPs across TNTs and seed new infections [Fig 8]. Future work to elucidate the exact mechanism of transport of the Rab11a-vRNP complexes, including the motors utilized and the fate of the incoming Rab11a-vRNP complexes in the recipient cytosol, are exciting avenues to be studied and will further our understanding of IAV-host interactions.

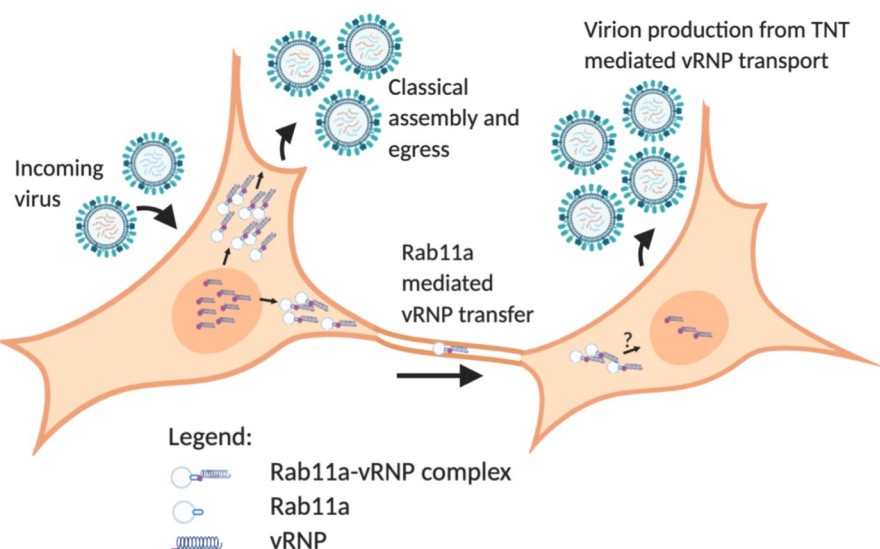

**Fig 8. Working model for Rab11a mediated vRNP transport across TNTs.** vRNP complexes synthesized within the nucleus are exported out and form Rab11a-vRNP complexes. Two potential fates of these complexes are shown- the classical assembly and egress pathway for production of progeny virions and transport of these complexes via TNTs to an uninfected cell. A new infection is initiated in the recipient cell, resulting in progeny virion production. Generated via BioRender.com.

## Materials and methods

### Cells and cell culture media

MDCK cells [obtained from Dr. Daniel Perez] and MDCK WSN HA cells [obtained from Dr. Ryan Langlois] were maintained Minimal Essential Medium [Sigma] supplemented with 10% fetal bovine serum [FBS; Atlanta Biologicals], penicillin [100 IUml$^{-1}$], and streptomycin [100 µg ml$^{-1}$; PS; Corning]. A549 WT, A549 Rab11a KO were maintained in Dulbecco's Modified Essential Medium [Gibco] supplemented with 10% FBS [Atlanta Biologicals], and PS. All cells were cultured at 37˚C and 5% $CO_2$ in a humidified incubator. All cell lines were tested monthly for mycoplasma contamination while in use. The medium for culture of IAV in each cell line [termed virus medium] was prepared by eliminating FBS and supplementing the appropriate medium with 4.3% BSA and PS. Ammonium chloride-containing virus medium was prepared by the addition of HEPES buffer and $NH_4Cl$ at final concentrations of 50 mM and 20 mM, respectively. OPTi-MEM [Gibco] was used as a serum free medium where indicated.

### Generation of Rab11aKO cells

Generation and characterization of Rab11a KO A549 cells wasreported in [12,46]. Briefly, two guide RNAs [gRNA] targeting the promoter and exon 1 of the Rab11a gene were used. Oligonucleotides for the CRISPR target sitesT1[forwardCACCGCATTTCGAGTAAATCGAGAC and reverseAAACGTCTCGATTTACTCGAAATGC] and T2 [forward CACCGTAACAT-CAGCGTAAGTCTCA and reverse AAACTGAGACTTACGCTGATGTTAC] were annealed and cloned into lentiCRISPRv2 [Addgene #52961] and LRG [Addgene #65656] expression vectors, respectively. A549 cells transduced with lentivirus vectors expressing gRNAs were selected in the presence of 2 µg/mL puromycin for 10 days and clonal Rab11a KO cells were generated by limiting dilution of the polyclonal population. Rab11a KO cells were identified by PCR analysis of the targeted genomic region using the following primers [forward

TGTTCAACCCCCTACCCCCATTC and reverseTGGAAGCAAACACCAGGAAGAACTC]
and further confirmed by western blot analysis of Rab11a expression [46].

## Viruses

All viruses used in this study were generated by reverse genetics [47]. For influenza A/Panama/2007/99 virus [P99; H3N2], 293T cells transfected with reverse-genetics plasmids 16–24 h previously were injected into the allantoic cavity of 9- to 11-d-old embryonated chicken eggs and incubated at 37˚C for 40–48 h. The resultant egg passage 1 stocks were used in experiments. For influenza A/Netherlands/602/2009 virus[NL09; pH1N1], 293T cells transfected with reverse-genetics plasmids 16–24 h previously were co-cultured with MDCK cells at 37˚C for 40–48 h. The supernatants were then propagated in MDCK cells at a low MOI to generate NL09 working stocks. The titers for these viruses were obtained by plaque assays on MDCK cells.

The NL09 ΔHA Venus WT, P99 ΔHA Venus WT and NL09 ΔHA Scarlet VAR viruses were generated by reverse genetics by co-culture with MDCK WSN HA cells rather than MDCK cells. The ΔHA Venus and ΔHA Scarlet rescue plasmids were prepared by inserting either the mVenus [48] or mScarlet [49] ORF within the HA sequence, retaining only the 3' terminal 136 nucleotides of the HA segment upstream of the reporter gene start codon and the 5' terminal 136 nucleotides of the HA segment downstream of the reporter gene stop codon. ATG sequences within the upstream portion were mutated to ATT to prevent premature translation start [50]. As previously described [51], one silent mutation was introduced into each NL09 cDNAto generate the NL09 VAR reverse genetics system, which was used to generate the NL09 ΔHA Scarlet VAR virus. These silent mutations enable differentiation of VAR virus segments from those of the WT virus using high-resolution melt analysis [27,52].

## Immunofluorescence and confocal imaging

For fixed cell imaging, MDCK, A549 WT or A549 Rab11a KO cells were seeded onto glass coverslips. Infection with either NL09 or P99 viruses was performed the next day by adding 250μl of inoculum to the coverslips and incubating at 37˚C for 1 h with intermittent rocking. Inoculum was removed, cells washed twice with 1X PBS and Opti-MEM added to the dish. After incubation at 37˚C for 24h,cells were washed with 1X PBS [Corning] thrice and fixed with 4% paraformaldehyde [AlfaAesar] for 15 minutes at room temperature. Cells were washed with 1X PBS and permeabilized using 1% Triton X-100 [Sigma] in PBS for 5 minutes at room temperature and washed with 1X PBS. Cells were stained with mouse anti NP antibody [Abcam ab43821][1:100], rabbit anti Rab11a antibody [Sigma HPA051697][1:100], and Phalloidin Alexa Fluor 647 [Invitrogen A22287][1:40] overnight at 4˚C. Cells were washed thrice with 1X PBS and incubated with donkeyanti mouse Alexa Fluor 555 [Invitrogen A32773][1:1000] and Anti rabbit Alexa Fluor 488 [Invitrogen A32731][1:1000] for 1 h at 37˚C. Coverslips were washed thrice with 1X PBS and mounted on glass slides using ProLong Diamond Anti-Fade Mountant with DAPI [Invitrogen P36962] prior to imaging.

Confocal images were collected using the Olympus FV1000 Inverted Microscope at 60X 1.49 NA Oil magnification on a Prior motorized stage. Images were acquired with a Hamamatsu Flash 4.0 sCMOS camera controlled with Olympus Fluoview v4.2 software. All images were processed using Fijiimage analysis software [53].

## Immunofluorescence and high-resolution STED imaging

A549 WT or A549 Rab11a KO cells were seeded onto glass coverslips. Infection with NL09 WT viruses was performed the next day by adding 250 μl of inoculum to the coverslips and

incubating at 37˚C for 1 h with intermittent rocking. Inoculum was removed, cells washed twice with 1X PBS and Opti-MEM added to the dish. After incubation at 37˚C for 24 h, cells were washed with 1X PBS [Corning] thrice and fixed with 4% paraformaldehyde [AlfaAesar] for 15 minutes at room temperature. Cells were washed with 1X PBS and permeabilized using 1% Triton X-100 [Sigma] in PBS for 5 minutes at room temperature and washed with 1X PBS. Cells were stained with mouse anti NP antibody [Abcam ab43821][1:50], rabbit anti Rab11a antibody [Proteintech 20229-1-AP][1:50], and CellMask Deep Red Actin Tracking Stain [Invitrogen A57248][1:1000] overnight at 4˚C. Cells were washed thrice with 1X PBS and incubated with goat anti- mouse STAR ORANGE [Abberior STORANGE-1001] and goat anti-rabbit STAR GREEN [Abberior STGREEN-1002] for 1 h at 37˚C. Coverslips were washed thrice with 1X PBS and mounted on glass slides using ProLong Diamond Anti-Fade Mountant with DAPI [Invitrogen P36962] prior to imaging.

For the A549 KO-WT co-culture experiment, WT cells were pre-stained with CellTracker Blue CMAC dye [Invitrogen C2110][1:500] overnight at 37˚C in OPTIMEM. This dye stains thiols in the cytoplasm and fluoresces blue under the DAPI filter. Cells were washed with 1X PBS, trypsinized and then mixed with NL09 infected Rab11a KO cells and seeded onto glass coverslips. After incubation at 37˚C for 24 h, cells were processed and stained as described above. Coverslips were washed thrice with 1X PBS and mounted on glass slides using ProLong Glass Anti-Fade Mountant [Invitrogen P36980] prior to imaging.

Images were taken using an Abberior Facility Line STED microscope, controlled via Lightbox. The microscope is a custom design with a built-inOlympus IX3-ZDC-12 z-drift compensation unit for steady confocal/STED imaging to correct for chromatic abberration. Single planes were acquired at 900 x 900 px [150 μm × 150 μm] with a 4μs dwell time and a line accumulation of 4. A 60X NA 1.4 [oil immersion; UPLXAPO60XO] objective lens was used with a confocal pinhole of 0.5 AU. The Alexa 647 STED channel [F-Actin] was excited with a 1mW pulsed 640nm laser line at 1% and spectral detection at 650–735 nm. 2D STED was induced with a 2750 mW 775nm pulsed laser at 25%. The STAR ORANGE channel [NP] was excited with a 200μW pulsed 561 nm laser line at 10% and spectral detection at 574–637 nm. 2D STED was induced with a 2750 mW 775 nm pulsed laser at 25%. The STAR GREEN channel [Rab11a] was excited with a 1mW pulsed485 nm laser line at 5% and spectral detection at 498–574 nm. 2D STED was induced with a 400mW 595 nm pulsed laser at 30%. The DAPI/CellTracker Blue channel was excited with a 20 mW 405 nm laser line at 3% and spectral detection at 415–497 nm. All 2D STED images were acquired at a 2 μs dwell time. All images were acquired using the same parameters and pixel intensities were consistent across all datasets.

## Quantification of co-localization

Co-localization efficiency of NP and Rab11a was calculated using the Manders coefficient [54] with the Costes threshold [55] on both A549 WT and Rab11a KO cells using the JaCoP plugin [56] in FiJi [53].The quantification of co-localization was limited to the TNT regions of interest. All images were acquired in 2D and quantification was performed on the single slice.

## Quantification of cell-cell transmission

MDCK, A549 WT and A549 KO cells were inoculated with NL09 ΔHA Venus WT or P99 ΔHA Venus WT virus at aMOI of 0.5 PFU/cell and incubated for 1 h at 37˚C. Cells were washed with 1X PBS [Corning] to remove residual inoculum and supplemented with OPTI-MEM [Gibco] without trypsin and in the presence of 30μM Cytochalasin D [Sigma] where indicated and incubated at 37˚C. Infected cells were counted manually by the presence of green fluorescence using an epifluorescence microscope [Zeiss] at the time points indicated

and binned into single infected or foci of infected cells. Foci were defined as clusters of at least two contiguous, positive, cells. The cell counts were graphed using the GraphPad Prism software [57].

## Co-culture for production of infectious virus

MDCK, A549 WT or A549 Rab11a KO cells were inoculated with either NL09 ΔHA Venus WT or NL09 ΔHA Venus VAR at a MOI of 25 PFU/cell or 2.5 PFU/cell and were incubated in virus medium without trypsin for 2 h at 37˚C. Cells were washed twice with 1X PBS [Corning] and then treated with PBS-HCl, pH 3.0 for 5 min at room temperature to remove residual inoculum. Cells were washed once with 1X PBS and then trypsinized using 0.5 M trypsin-EDTA [Corning]. Cell slurry was collected in growth medium containing FBS and centrifuged at 1000 rpm or 5 minutes in a tabletop centrifuge [ThermoSorvall ST16]to pellet cells. Supernatant was aspirated and cells resuspended in virus medium containing TPCK-trypsin [Sigma], 20 mM HEPES [Corning] and 50 mM $NH_4Cl$ [Sigma] with or without 30 μM Cytochalasin D [Sigma] or 30 μM Nocodazole [Sigma] as indicated. Infected cell slurry was mixed with naïve MDCK WSN HA cells in a ratio of 1:1:2 of NL09 ΔHA Venus WT: NL09 ΔHA Scarlet VAR: Naïve MDCK WSN HA cells respectively and plated onto 6-well plates. Cells were allowed to attach at 37˚C and supernatant was collected at indicated time points for analysis.

## Conventional production of infectious virus with Cytochalasin D treatment

MDCK cells were infected with NL09 WT virus at a MOI of 25 PFU/cell, acid washed as described above to remove residual inoculum and incubated with or without 30 μM Cytochalasin D in virus medium containing TPCK-trypsin. Supernatant was collected at indicated time points for analysis.

## Quantification of reassortment

Reassortment was quantified for coinfection supernatants as described previously [27]. Briefly, plaque assays were performed on MDCK WSN HA cells in 10cm dishes to isolate virus clones. Serological pipettes [1 ml] were used to collect agar plugs into 160 μl PBS. Using a ZR-96 viral RNA kit [Zymo], RNA was extracted from the agar plugs and eluted in 40 μl nuclease-free water [Invitrogen]. Reverse transcription was performed using Maxima reverse transcriptase [RT; ThermoFisher] according to the manufacturer's protocol. The resulting cDNA was diluted 1:4 in nuclease-free water, each cDNA was combined with segment-specific primers [S1 Table] and Precision melt supermix[Bio-Rad] and analysed by qPCR using a CFX384 Touch real-time PCR detection system [Bio-Rad] designed to amplify a region of approximately 100 bp from each gene segment that contains a single nucleotide change in the VAR virus. The qPCR was followed by high-resolution melt analysis to differentiate the WT and VAR amplicons [27]. Precision Melt Analysis software [Bio-Rad] was used to determine the parental virus origin of each gene segment based on the melting properties of the cDNA fragments and comparison to WT and VAR controls. Each plaque was assigned a genotype based on the combination of WT and VAR genome segments, with two variants on each of seven segments allowing for 128 potential genotypes.

## Supporting information

**S1 Fig. IAV vRNPs associate with Rab11a within F-Actin rich TNTs in MDCK cells.** MDCK cells were mock-infected or infected with NL09 or P99 viruses. Cells were stained for

DAPI [blue], NP [red], Rab11a [green] and F-Actin [pink]. Scale bar is 20μm for all images.
(TIF)

**S2 Fig. IAV vRNPs associate with Rab11a within F-Actin rich TNTs in A549 WT cells.** A549
WT cells were mock-infected or infected with NL09 or P99 viruses. Cells were stained for DAPI
[blue], NP [red], Rab11a [green] and F-Actin [pink]. Scale bar is 20μm for all images.
(TIF)

**S3 Fig. IAV vRNPs associate with Rab11a within F-Actin rich TNTs in A549 Rab11a KO
cells.** A549 Rab11a KO cells were mock-infected or infected with NL09 or P99 viruses. Cells
were stained for DAPI [blue], NP [red], Rab11a [green] and F-Actin [pink]. Scale bar is 20μm
for all images.
(TIF)

**S1 Table. Sequence specific primers for High Resolution Melt Analysis.** Forward and
Reverse primer sequences for all eight genome segments from NL09 [PB2, PB1, PA, HA, NP,
NA, M and NS] are depicted in the table.
(XLSX)

## Acknowledgments

We thank Drs. Daniel Perez and Ryan Langlois for generous sharing of cell lines. We also
thank the Emory University Integrated Cellular Imaging Microscopy Core for assistance in
image acquisition. The content of this article is solely the responsibility of the authors and does
not necessarily represent the official views of the National Institutes of Health.

## Author Contributions

**Conceptualization:** Ketaki Ganti.

**Data curation:** Ketaki Ganti, Anice C. Lowen.

**Formal analysis:** Ketaki Ganti, Anice C. Lowen.

**Funding acquisition:** Anice C. Lowen.

**Investigation:** Ketaki Ganti, Anice C. Lowen.

**Methodology:** Ketaki Ganti, Anice C. Lowen.

**Project administration:** Anice C. Lowen.

**Resources:** Ketaki Ganti, Julianna Han, Balaji Manicassamy, Anice C. Lowen.

**Software:** Ketaki Ganti.

**Supervision:** Balaji Manicassamy, Anice C. Lowen.

**Validation:** Ketaki Ganti.

**Visualization:** Ketaki Ganti, Anice C. Lowen.

**Writing – original draft:** Ketaki Ganti.

**Writing – review & editing:** Julianna Han, Balaji Manicassamy, Anice C. Lowen.

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
