## [Decision Letter · Decision Letter 0]

14 Feb 2021

Dear Dr Ganti,

Thank you very much for submitting your manuscript "Rab11a mediates cell-cell spread and reassortment of influenza A virus genomes via tunneling nanotubes." for consideration at PLOS Pathogens. As with all papers reviewed by the journal, your manuscript was reviewed by members of the editorial board and by several independent reviewers. In light of the reviews (below this email), we would like to invite the resubmission of a significantly-revised version that takes into account the reviewers' comments.

The proposed model Rab11a-mediated cell-to-cell spread via nanotubes is a novel and exciting mechanism for influenza A virus dissemination. However, all three reviewers had significant reservations as to whether the experiments used in this study fully support this model. I therefore have to ask for major revisions. Additional work needs to include the major points voiced by reviewers, with a focus on: 1. enhancing image quality and quantify imaging data, 2. discussing limitations of experimental systems and alternative explanations of findings, 3. addressing specific comments as to additional control experiments voiced by reviewer 1 and 3. Although experiments in air-liquid interface as requested by reviewer 2 would be nice, I acknowledge that these experiments would also be experimentally challenging. Thus, I recommend clearly discussing the limitations of A549 or other monolayer cultures used in this study.

We cannot make any decision about publication until we have seen the revised manuscript and your response to the reviewers' comments. Your revised manuscript is also likely to be sent to reviewers for further evaluation.

Sincerely,

Meike Dittmann, Ph.D.

Associate Editor

PLOS Pathogens

Carolina Lopez

Section Editor

PLOS Pathogens

Kasturi Haldar

Editor-in-Chief

PLOS Pathogens

orcid.org/0000-0001-5065-158X

Michael Malim

Editor-in-Chief

PLOS Pathogens

orcid.org/0000-0002-7699-2064

The proposed model Rab11a-mediated cell-to-cell spread via nanotubes is a novel and exciting mechanism for influenza A virus dissemination. However, all three reviewers had significant reservations as to whether the experiments used in this study fully support this model. I therefore have to ask for major revisions. Additional work needs to include the major points voiced by reviewers, with a focus on: 1. enhancing image quality and quantify imaging data, 2. better discussion of limitations of experimental systems and alternative explanations of findings, 3. specific comments as to additional control experiments voiced by reviewer 1 and 3. Although experiments in air-liquid interface would be nice, I acknowledge that these experiments would also be experimentally challenging. Thus, I recommend clearly discussing the limitations of A549 or other monolayer cultures used in this study.

Reviewer's Responses to Questions

**Part I - Summary**

Reviewer #1: In the present manuscript, Ganti et al. extend previous observations that IAV can spread through TNTs to include a role for Rab11a (already known to facilitate RNP trafficking to the plasma membrane) in mediating genome transport directly between cells.

The authors have established and effectively utilized several very nice experimental systems to dissect viral spread through TNTs, including HA-deficient reporter viruses harboring genetic tags (paired with an HA-expressing ‘recipient’ cells) and Rab11a KO cells. Still, the paper is heavily based on selected images, lacks discussion of the literature on Rab11a’s role in TNT formation and trafficking in other contexts, and fails to present convincing data that Rab11a mediates bidirectional transport of viral genomes, weakening their overall conclusions. Overall, tempering their conclusions and improving analysis by enhancing image quality and/or pairing their microscopy with more quantitative approaches would improve the paper.

Reviewer #2: In the manuscript entitled, “Rab11a mediates cell-cell spread and reassortment of influenza A virus genomes via tunneling nanotubes”, the authors address whether tunneling nanotubes (TNTs) facilitate an alternative cell-spread mechanism during influenza infection. Using primarily static imaging techniques, Rab11A knockout cells, mutant viruses, and drug treatments the authors conclude that Rab11a transports NP (presumably viral RNP complexes – although viral RNA was not specifically visualized), through TNTs to neighboring cells. The author cleverly utilize an HA-deficient viruses, which cannot undergo secondary infection through a receptor-mediated process, to control for HA-dependent spread between cells. Capitalizing on the VAR virus strategy from the Lowen lab, the authors observe a reduction in reassortment events infected Rab11a KO cells compared to WT cells in a coinfection experiment, which is the most exciting result of this paper but unfortunately is not followed up.

The use of TNTs in influenza assembly is an exciting concept. Unfortunately, this study falls short of conclusively demonstrating that TNTs are used for transport of vRNP. The following experiments are suggested to provide a stronger foundation for which to conclude that TNTs are involved.

Reviewer #3: This study by Ganti et al. extends previous work showing that influenza virus uses intercellular nanotubes for cell-to-cell spread of their segmented genome (in the form of viral ribonucleoproteins or vRNPs), by investigating to what extent the host GTPase Rab11 is involved in this process. The experimental approach is innovative in two respect : i) HA-deficient reporter viruses were used, that are unable to produce of infectious progeny virions at the plasma membrane (instead of antibodies or drugs interfering with secondary infection in previous studies), and ii) genetically tagged viruses were used to investigate the potential of TNTs to mediate genetic mixing and reassortment.

The findings are novel and will be of interest for readers in the field. In particular, the observed difference between Rab11+ and Rab11-KO cells in richness of viral genotypes produced in a TNT-dependent experimental setting, is in favor of a role of Rab11 in TNT-mediated spread of influenza viruses. However to fully support the conclusions and model proposed by the authors, some experiments need to be strengthened with additional quantifications and/or controls.

**Part II – Major Issues: Key Experiments Required for Acceptance**

Reviewer #1: 1) The authors comment on colocalization of Rab11a with NP inside TNTs in Fig 1, yet this is very difficult to discern from the images provided. NP appears to fill the entire cytosol. Higher magnification / better resolution of the TNTs in these images would be helpful. Also, colocalization analysis would strengthen their argument (e.g. does 100% of NP within TNTs colocalize with Rab11a?). Note that one of the images in the main figure for the KO cells appears to be duplicated in the supplemental figure (S1C).

2) As no characterization of the KO cells is provided beyond the lack of Rab11a staining in IF images, a rescue experiment would provide confirmation that the phenotype (loss of foci formation) is indeed specific for Rab11a.

3) The authors suggest that the production of infectious virus in Fig 4C could be due to the action of Rab11a from the HA-expressing recipient cells. To support this, they coculture A549 cells expressing Rab11a-mCherry with IAV-infected Rab11a KO cells and note that they “observe colocalization of Rab11a and NP in the Rab KO cells, indicating the transfer of mCherry Rab11a to the infected cells.” How do we know that this is Rab11a-mCherry in KO cells and not NP transfer to Rab-mCherry expressing cells? Overall these data are not sufficient to support the conclusion that TNTs allow bidirectional movement and that Rab11a from uninfected cells can essentially grab RNPs from their neighbors.

4) Based on their data showing low efficiency, it seems unlikely that all 8 segments are routinely transported from cell-cell (at least in this system). Indeed, the authors state in the conclusion that “…indicates that vRNPs may be trafficked individually or as subgroups and not as a constellation of 8 segments”. Thus, do the authors think this is a true driver of virus spread and means to evade immune detection? How do the authors think TNTs play a role in infection in the airway? Is this likely between airway cells and other cells types? More discussion on this topic is warranted to better understand the relevance of these findings during natural infection especially since the cell lines and culture conditions used do not recapitulate the tightly packed, polarized epithelial cells in vivo.

5) At line 222, the authors speculate that fewer TNT connections may underlie the fact that fewer reassortant viruses are observed when Rab11a is absent from infected donor cells. This argument could be bolstered by quantitation of these connections, and/or citing the literature on this topic (e.g. DOI: 10.1242/jcs.215889 ; DOI: 10.1038/cddis.2016.441). If mutation of either in Rab11a or PB2 to prevent the direct interaction is possible, it may provide a means to decouple the role of Rab11a in TNT formation (indirect) vs. RNP trafficking (direct). At the moment, it seems assumed that Rab11a is responsible for RNP trafficking through the TNTs based on a similar role for Rab11a in bringing RNPs to the site of assembly and the images in Fig 1 (see point #1 above)).

Reviewer #2: Additional Experiments:

1. Examine cell-to-cell spread through TNTs in differentiated airways cells at an air liquid interface. The studies presented are restricted to A549 cells sparsely seeded on the coverslip to create the extended filaments between the cells. However, in the airway cells are densely packed together with TNTs are found through tight junctions between cells; thus, a physiologically relevant system should be used to complement the studies presented.

2. Image TNTs and the cargo (NP and vRNP) within them using super-resolution or electron microscopy to confirm that they are within the tube and not traveling along the outside of the plasma membrane.

3. Examine the role of microtubules on HA-independent cell-to-cell spread. The authors focus on the use of F-actin in TNTs and use Cytochalsin D (actin depolymerization durg) to examine cell-to-cell spread. However, the authors should include a microtubule depolymerization treatment as a control to ensure that the impact on cell-to-cell spread is dependent upon actin and not just on the disruption of cytoskeletal trafficking.

Other Major Concerns:

1. Details on image acquisition and analysis are missing. It is unclear if authors used computational or manual image analysis pipelines. What were the thresholds used for the analysis etc. It is difficult to interpret the data without these details.

2. The authors ignore many caveats to the experimental setup and alternative entry pathways, such as:

a. Rab11a has a multitude of roles in intercellular membrane trafficking of cellular and viral components. Presumably, the canonical pathway of IAV assembly would have been disrupted in the knockout system many steps upstream of vRNPs reaching the TNTs.

b. Depolymerization of actin drastically impacts cell morphology that could impact trafficking of vRNP.

c. There may be other strategies of cell-to-cell spread that are HA-independent in addition to TNTs.

d. The experiments leading to the conclusion of bidirectional Rab11A should be done with live cell imaging, since in the current model the Rab11A-mCherry cells could be infected with virus produced in the Rab11A KO cells.

3. In addition, the viral titers in Fig 4C, suggest to this reviewer that Rab11A is unnecessary for influenza assembly. It is surprising that the authors conclude that viral replication between the WT and KO cells was significantly different when less than a 0.5 log10 decrease was observed. The authors need to resolve why Rab11A would be necessary for vRNP TNT spread and endocytic-based assembly pathways, but not dramatically impact viral titers.

Reviewer #3: Figure 1. In addition to representative images, quantitative data should be provided to to convincingly support the authors' conclusions: how many TNTs were observed in each conditions ? what were the proportions of NP+ Rab11+, NP+ Rab11- and NP-Rab11+ TNTs ?

From a mechanistic perspective it would also be of interest to clarify the following points: in mock-infected cells, were TNTs positive for Rab11 staining, i.e. are Rab11-positive vesicles naturally trafficking through TNTs ? was the average number of TNTs per cell enhanced upon viral infection as described previously by Roberts et al (ref # xx) ? was the average number of TNTs per cell modified by Rab11 KO ?

Finally, it would be preferable to state that infected cells show TNTs that are positive for both Rab11 and NP staining, as the imaging resolution is not sufficient to demonstrate colocalisation of the two proteins, and therefore does not allow to conclude that “vRNPs associate with Rab11a” within TNTs (line 95), although this association is likely.

Figure 4. As high MOIs are used for the initial infection and low infectious titers (~10e3-10e4 PFU/ml) are detected in the supernatant of the coculture, it is important to check that there is no residual infectivity upon acid wash. The percentages of Venus-positive and Scarlet-positive cells in the coculture should be provided to allow a thorough interpretation of the data showed in Figure 4 and Figure 6.

Figure 5. Figure 5 does not convincingly demonstrate that Rab11 transfer from non-infected cells to infected cells could account for TNT-dependent cell-to-cell spread in the experiment shown in Figure 4B. In DMSO cells, could NP+ mCherryRab11+ TNTs be observed ? if yes, was the presence of NP in the TNTs strictly dependent upon the presence of mCherry-Rab11 ? if no, can NP+ mCherryRab11+ TNTs be observed at later time-points ?

To further confirm the author’s hypothesis of Rab11 transfer from non-infected cells to infected cells, the experiment shown in Figure 4B should be performed using MDCK-WSN-HA cells depleted in Rab11a, eg pre-treated with siRNAs targeting Rab11a.

**Part III – Minor Issues: Editorial and Data Presentation Modifications**

Reviewer #1: 1) Figure 6 would benefit from additional labeling within the figure itself

2) Line 58 should be either “comprises eight RNA” or “is composed of”

3) Line 97 seems to be missing a reference after the first sentence of the results “Upon nuclear exit…”

4) Line 144 should read “shown in Figure 3”

Reviewer #2: 1. All images presented in this manuscript are small include a wide field of view. They would benefit from zoomed-in insets of regions of interests, highlighted interactions within the tunneling nanotubes, and a scale bar needs to be added.

2. Line 144 – missing “in” before “figure 3, this is…”.

3. Fig. 6 lacks the “A” and “B” subpanel labels mentioned in the text.

Reviewer #3: Figure 2 and 3. The effects of cytochalasin or Rab11-KO are quite striking, in that they suppress the increase in foci number over time. However the authors should show a baseline i.e. the same counts at an earlier time point such as 4-6 hpi, to demonstrate that the ~50 foci detected in the presence of cytochalasin or Rab11-KO were most likely formed during the initial infection round. It would also be interesting to show analysis of the same experiments with a higher threshold of ≥3 cells instead of ≥ 2 cells per foci.

Figure legends. Indications regarding MOI and time-point should be provided in the main text and/or legend figure for each experiment. Line 534 and line 540, the number of independent experiments should be indicated. Line 549 : whether the three replicates are technical or biological replicates should be clarified.

In the introduction, line 60-68. When referring to the role of Rab11 in the trafficking of newly synthesized vRNPs, the recent review by M Amorim (PMID 30687703) should be cited in addition to ref #2. Recent mechanistic insights into the transport of Rab11-vRNP complexes provided by the studies of Alenquer et al (PMID 30967547) and de Castro Martin et al (PMID 29123131) should be cited in addition to ref #5-8.

Line 96. Appropriate references (PMID 21307188 and 23063830) should be cited to support the notion that vRNPs bind to Rab11 via PB2.

Line 248. The manuscript provides no evidence that loss of Rab11a leads to a more “dispersed NP localisation within the cytoplasm”. The sentence should be rephrased.

Line 51. The manuscript provides no evidence that “complexes of Rab11a and viral components can be trafficked” across tunelling nanotubes. loss of Rab11a leads to a more “dispersed NP localisation within the cytoplasm”. The sentence should be rephrased.

PLOS authors have the option to publish the peer review history of their article (what does this mean?). If published, this will include your full peer review and any attached files.

Reviewer #1: No

Reviewer #2: No

Reviewer #3: No
---

## [Decision Letter · Decision Letter 1]

9 Aug 2021

Dear Dr Ganti,

Thank you very much for submitting your manuscript "Rab11a mediates cell-cell spread and reassortment of influenza A virus genomes via tunneling nanotubes." for consideration at PLOS Pathogens. As with all papers reviewed by the journal, your manuscript was reviewed by members of the editorial board and by several independent reviewers. The reviewers appreciated the attention to an important topic and found this revision 1 much improved from the original submission. Based on the reviews, we are likely to accept this manuscript for publication, providing that you modify the manuscript according to the review recommendations, with no additional experiments needed. 

Sincerely,

Meike Dittmann, Ph.D.

Associate Editor

PLOS Pathogens

Carolina Lopez

Section Editor

PLOS Pathogens

Kasturi Haldar

Editor-in-Chief

PLOS Pathogens

orcid.org/0000-0001-5065-158X

Michael Malim

Editor-in-Chief

PLOS Pathogens

orcid.org/0000-0002-7699-2064

All three reviewers found this revision much improved from the original submission. Therefore, I recommend minor revision with no additional experiments needed.

Reviewer Comments (if any, and for reference):

Reviewer's Responses to Questions

**Part I - Summary**

Reviewer #1: The authors have submitted a revised version of their study which suggests Rab11a mediates vRNP transport through TNTs to promote cell-cell spread and genome reassortment. Overall, the incorporation of new data (including high resolution microscopy of NP+/Rab11a+ TNTs and colocalization analysis) has improved the manuscript and strengthened their conclusions.

Notably, the data in Figure 6 are still not definitive proof that these TNTs are bidirectional. (The image provided suggests Rab11a is actually not required for vRNPs to enter TNTs. Furthermore, we cannot know which direction the NP+/Rab11a+ puncta are going from a still image.) Live cell imaging would really be needed to address this hypothesis. However, the authors are careful in their discussion of these findings (e.g. lines 312-316), thus, I am satisfied with their updated approach.

Reviewer #2: In the revised manuscript entitled, “Rab11a mediates cell-cell spread and reassortment of influenza A virus genomes via tunneling nanotubes”, the authors have addressed many of the original concerns by including super resolution microscopy and quantitation of Rab11a and NP colocalization. The authors now conclusively demonstrate that in A549 cells the TNT are formed that contain both Rab11a and IAV NP. The use of HA-deficient viruses with a co-culture system of wt and HA-expressing cells reveals a strong role for actin in HA-independent spread of the virus. As mentioned in the original review one of the most compelling observations is that the reduction in reassortment events infected Rab11a KO cells compared to WT cells with the VAR and WT coinfection studies.

The authors have improved upon the first version of this manuscript and modification of the discussion, figures, methods and some conclusions are requested to avoid over-interpretation of the results.

Reviewer #3: In the revised version of the manuscript, the authors have adressed most of the concerns raised by the reviewers. A few points should still be improved.

**Part II – Major Issues: Key Experiments Required for Acceptance**

Reviewer #1: (No Response)

Reviewer #2: (No Response)

Reviewer #3: (No Response)

**Part III – Minor Issues: Editorial and Data Presentation Modifications**

Reviewer #1: -The cells in Figure 2A should be labeled as Rab11a KO in the figure for clarity

-Line 157 – I believe the authors mean Venus-positive cells as these are HA deleted viruses here.

Reviewer #2: 1. Since tight junctions could function similarly to TNT in differentiated airways and other viruses (hMPV and measles viruses – see work by Dutch Lab and Cattaneo Lab) have been shown to travel cell-cell through these sort of connections it is feasible IAV would as well. Inclusion of this point in paragraph starting on line 334 would be nice.

2. STED imaging comments:

a. Was there any correction for chromatic aberration?

b. What was considered in “close proximity”? An image resolution of 30 nm can provide a more precise quantification for colocalization between Rab11a and NP is possible in an unbiases computational manner (rather than manual counting).

c. Was a 30nm resolution confirmed for your specific images and setup? If not – please provide the limit of resolution

d. In Fig 2 and 6 the resolution of the high resolution images is still pixelated when zoomed in. Please consider making the zoomed images larger so that the power of the STED images can be revealed.

3. Fig 1 – no scale bars is included. The legend of the cells is inaccurate. The resolution and image size makes it difficult to assess colocalization.

4. Fig 2 – what is the time point of infection? Was the mander’s colocalization coefficient used only on TNT region of interest? Please include that detail in the methods. Also, was the coefficient derived from a single slice or an 3D volume of the cell? Please include these details in the methods.

5. Fig 6 – include the number of independent replicates for this study. What was the time post infection. The Rab11a signal is interested since it is present in the cytoplasm of the KO cell as well. Please comment on whether the same staining procedures, imaging parameters and pixel intensities are consistent between this data set and the STED images in Fig 2 of Rab11a KO cells.

6. Please note that STED images do not indicate direct interaction. The examination of PB2 and Rab11a direct interaction is most convincingly shown in Avilov et al – using bi-fluorescence complementation assay with split GFP on PB2 and Rab11a. Consider modifying line 122.

Reviewer #3: In the absence of live imaging data, the authors’ claim that they find evidence for bidirectional movement of Rab11 through TNTs (e.g line 226, lines 276-277) should be attenuated. The experiment shown in Figure 6 does not prove bi-directionality. The only argument for bidirectionality is the production of viral progeny when A549-Rab11KO cells are co-cultured with MDCK-HA cells (Figure 5E), however it is a very indirect argument. Therefore bidirectionality may be referred to as likely, but not certain.

Line 221: it should be clarified “with a marginal difference in titers at 48 and 72 hpi ”, as the difference in titers between A549-WT and A549-Rab11KO cells is not marginal at 24 hpi.

Figure 2 : the inclusion of STED imaging and quantification data represent a significant improvement. However the confocal and STED images shown in Figure 2A are very pixelated, and higher quality images should be provided (the same comment applies to Figure 6). The labeling of the right panel of Figure 2A should state « Rab11a-KO ».

In Figure 2B, does it really make sense to quantify the co-localisation of NP with Rab11 in Rab11-KO cells ?

In the introduction, lines 65-71: the notion that an intact microtubule network is important for the transport of vRNP should be tempered, as several pulbications have reported only a modest decrease of the production of infectious viruses upon treatment with nocodazole. The authors themselves, in Figure 5D of the present manuscript, show no effect (or even a positive effect) of nocodazole treatment on the productious of infectious viruses.

PLOS authors have the option to publish the peer review history of their article (what does this mean?). If published, this will include your full peer review and any attached files.

Reviewer #1: No

Reviewer #2: No

Reviewer #3: No

Figure Files:

Data Requirements:

Reproducibility:

References:

---

## [Editor Report · Decision Letter 2]

22 Aug 2021

Dear Dr Ganti,

We are pleased to inform you that your manuscript 'Rab11a mediates cell-cell spread and reassortment of influenza A virus genomes via tunneling nanotubes.' has been provisionally accepted for publication in PLOS Pathogens.

Best regards,

Meike Dittmann, Ph.D.

Associate Editor

PLOS Pathogens

Carolina Lopez

Section Editor

PLOS Pathogens

Kasturi Haldar

Editor-in-Chief

PLOS Pathogens

orcid.org/0000-0001-5065-158X

Michael Malim

Editor-in-Chief

PLOS Pathogens

orcid.org/0000-0002-7699-2064
---

## [Editor Report · Acceptance letter]

27 Aug 2021

Dear Dr Ganti,

We are delighted to inform you that your manuscript, "Rab11a mediates cell-cell spread and reassortment of influenza A virus genomes via tunneling nanotubes.," has been formally accepted for publication in PLOS Pathogens.

Best regards,

Kasturi Haldar

Editor-in-Chief

PLOS Pathogens

orcid.org/0000-0001-5065-158X

Michael Malim

Editor-in-Chief

PLOS Pathogens

orcid.org/0000-0002-7699-2064